# Comprehensive Evaluation of NIMBY Phenomenon with Fuzzy Analytic Hierarchy Process and Radar Chart

**Jian Wu** [1,2], **Ziyu Wang** [1] , **Xiaochun Bai** [2] **and Nana Duan** [1,*]

1   State Key Laboratory of Electrical Insulation and Power Equipment, Xi'an Jiaotong University, Xi'an 710049, China; 3122104010@stu.xjtu.edu.cn (Z.W.)
2   Shaanxi Electric Power Research Institute of State Grid Corporation of China, Xi'an 710000, China
*   Correspondence: duannana@mail.xjtu.edu.cn; Tel.: +86-029-8266-8623

**Abstract:** The risk level of the NIMBY (Not In My Back Yard) phenomenon is crucial for the safety and economy of transmission and transformation projects which is rarely studied, especially for site selection and the construction of transmission lines and substations. In order to effectively evaluate the risk level to solve the dilemma caused by the NIMBY phenomenon, an evaluation method for quantifying the level of the NIMBY phenomenon is proposed. In this paper, thirty-one evaluation criteria and a risk model are put forward according to relevant laws and regulations that should be followed in the transmission and transformation project in China, then the scores corresponding to these criteria are obtained by a questionnaire survey. The radar chart method and minimum area method are applied to determine the weights of the element and unit layers. Furthermore, the overall risk level is evaluated by the fuzzy comprehensive evaluation method. In addition, a transmission and transformation project in Xi'an City, China, is used as an example to verify the correction of the risk model and its evaluation method. The results show that the weaknesses in the transmission and transformation project are analyzed, and suggestions for decreasing the risk level are put forward to minimize losses due to the NIMBY phenomenon.

**Keywords:** transmission and transformation project; NIMBY; risk-level comprehensive evaluation; fuzzy analytic hierarchy process (FAHP); radar chart

## 1. Introduction

Risk evaluation is critical to the development of various industries, particularly the electric power industry. The electric power industry is an important basic energy industry in the development of national economy. A safe and reliable power network promotes the development of the country's economy and the well-being of its people. At present, countries all over the world are in a phase of such rapid development that the number and scale of power constructions required are increasing, and likewise, the capital and labor costs consumed are enormous. Therefore, reducing risk-assessment levels for the electric power industry helps to reduce construction costs, thereby contributing to the long-term growth and reliability of the business [1]. The risk evaluation of the electric power industry is, thus, of vital importance to people, power grid corporations, society, the economy and national security. A safe, reliable and sustainable electrical power system can provide people with a more convenient and comfortable life.

However, in order to ensure the reliability and quality of the power supply, the substation construction of the electrical power system must be deep in the load center, which is densely populated, and its land must be highly developed. In addition, the visual intrusion, noise and potential health risks of the electromagnetic fields emitted by electromagnetic devices are burdens [2]. There are two main sources of noise: one is the corona discharge of transmission lines, and the other is the vibration of transformer cores and windings due to the electromagnetic force generated by alternating magnetic fields [3]. Meanwhile, electric-power equipment generates electromagnetic radiation, which interferes with nearby radio

equipment, televisions, telephones, etc., causing image distortion, sound noise or communication interruption. Therefore, the siting of power lines and substations has often been met with public opposition, leading to significant planning delays and financial losses. The construction of transmission and transformation projects in the large-scale urban core area has become a common problem plagued by the power-grid corporations. Such opposition due to the public's misunderstanding of the effects of electric-power facilities is commonly labeled *NIMBY*, meaning the public exhibit a *not-in-my-backyard* behavior. It is defined interpreted by Dear in 1992 as "the protectionist attitudes and oppositional tactics of community groups in the face of unwelcome development in their neighborhoods [4]". Initially, the NIMBY behavior is generally viewed as a negative, showing that people prioritize their personal interests over the interests of society [5]. As more and more siting events occur as a result of the NIMBY phenomenon, some academics have critiqued the NIMBY explanation as shortsighted, pointing out that canned explanations mask the real reasons why people support or oppose wind-farm development [6–8]. In addition, some scholars provide a detailed review of the work of their predecessors. Based on the thirty years of North American wind-energy acceptance research, Rand et al. [9] conclude that the socioeconomic impacts of wind development are strongly tied to acceptance and issues of fairness, participation and trust during the development process influence acceptance. In order to lay the necessary groundwork of the NIMBY phenomenon for synthesis attempts, Borell et al. [10] critically scrutinize and analyze the first wave of research, which is a series of attitudinal data, and contrast later approaches which reflect the real-life protests against the establishment of human services.

The NIMBY phenomenon refers to the fear of residents or local units that construction projects, such as transmission and transformation projects, waste-disposal plants, nuclear-power plants and funeral parlors, will bring about a number of negative impacts on physical health, environmental quality and asset value, thus stimulating people's dislike of the complex and fostering the mentality of 'do not build it in my backyard', which means they will take a strongly collective opposition and even demonstrate resistance behavior [11–13].

Research on the NIMBY phenomenon in the world focuses on the construction of waste-disposal plants, nuclear power plants and funeral parlors, and seldom deals with the aspects of transmission and transformation projects. Moreover, the planning and construction of transmission and transformation projects are highly different from those mentioned above, which are mainly reflected in the planning of power grids, construction periods, line direction, substation-layout locations and social-demand levels. In addition, the little research on the NIMBY phenomenon of transmission and transformation projects are mostly approached from the aspects of project management, technological innovation and economic compensation, and seldom explores it from the direction of analyzing the existing cases, researching the relationship between the relevant stakeholders and establishing a risk model to solve the problem [14–17]. For example, Devine [18] discusses the factors affecting the siting problem of the 80 km 400 kV Lackenby–Picton–Shipton line in the northeast of England through questionnaires and hierarchical linear regression analysis. Bidwell [19] theoretically explored the impact of general values and beliefs on wind energy. Song et al. [20] proposed the genetic algorithms to minimize the total degree of NIMBY sentiments.

For the risk assessment of transmission and transformation projects, the impact variables are qualitative, not quantitative. Therefore, the use of fuzzy comprehensive evaluation is extremely appropriate. Meanwhile, the fuzzy comprehensive evaluation refers to the use of fuzzy mathematics methods to assess the possibility of evaluating fuzzy objects influenced by multiple factors based on certain evaluation criteria [21–23]. The method has been widely used in comprehensive evaluation in various fields and has achieved good results [24–26].

A risk assessment is carried out with reference to a number of evaluation indicators, which are assigned different weights depending on their level of importance. The weight

calculation is generally divided into subjective and objective weights, because the evaluation indicators of the risk assessment are qualitative, so the subjective weights like the analytic hierarchy process (AHP) method is commonly used as a useful multiple-criteria decision-making tool or a weight-estimation technique in various areas of human needs and interests [27]. The conventional AHP is a simple and explicit method to acquire a judgement matrix by exact ratios or numbers. However, due to the complexity, vagueness and uncertainty involved in real-world decision-making problems, decision makers discuss how the fuzzy judgement is more practical than explicit comparisons [28]. Nowadays, the combination of the conventional AHP and fuzzy methods to form a fuzzy AHP has been widely used. The method, called the fuzzy analytic hierarchy process (FAHP), first introduced by Van Laarhoven and Pedrycz, is a powerful tool to treat uncertainty in the case of incomplete or vague information [29]. Patil et al. [30] proposed a fuzzy AHP-TOPSIS framework for ranking the solutions of knowledge-management adoption. Mosadeghi et al. [31] compared the fuzzy AHP and AHP in an urban land-use planning decision and proposed to choose different methods depending on the complexity of the planning. Jiang et al. [32] used the fuzzy comprehensive evaluation method to solve the problem of the resource evaluation of a port shoreline.

However, the above method can not enable visualization to reflect certain characteristics of high-dimensional data or the overall relationship between data. The radar chart method is a multi-variable analytical method for comprehensive evaluation that can visually reflect the relative strengths and weaknesses of the evaluation system on the trend of each indicator [33]. Wang et al. [34] combined the AHP and entropy weight method based on the radar chart method to solve the comprehensive evaluation problem of power transmission and transformation projects. Wang et al. [35] used the fuzzy analytic hierarchy process and improved radar method to comprehensively evaluate and rank the braking performance of three types of hydrodynamic retarders.

Therefore, in order to solve the dilemma caused by the NIMBY phenomenon of transmission and transformation projects, a fuzzy analytic hierarchy process method based on improved an radar chart is suggested. For this aim, six units and thirty-one elements are proposed for constructing a hierarchical structure of performance evaluation. Then, the expert evaluation method, the radar chart method and the minimum area method are used to determine the weights of the element and unit layers. Finally, a transmission and transformation project in Xi'an City, China, is used as an example to verify the correction of the risk model and its evaluation method. The detailed contributions and novelty of this paper are as follows:

(1) Based on the laws and regulations that should be followed in the transmission and transformation project in China, a risk-evaluation index system of the NIMBY phenomenon that can accurately assess the risk model of transmission and transformation project is derived;

(2) The fuzzy analytic hierarchy and radar chart methods are applied to calculate the weights of the evaluation criteria. The imprecise decision maker's judgments are represented as fuzzy numbers rather than exact numerical values;

(3) The proposed method is feasible and effective, which has been verified by the transmission and transformation project in Xi'an City. It further provides suggestions based on the research results for risk evaluation and serves as a reference for future research in this field.

## 2. Risk Evaluation System of NIMBY Phenomenon

### 2.1. Evaluation Process

The selection of evaluation indicators is a crucial step in the risk evaluation of the NIMBY phenomenon, as it directly affects the evaluation results. The risk evaluation indicators should follow the principles of systematicity, scientific quality, comparability and practicability [36–38]. In addition, each of these principles needs to be reflected in the

evaluation results through a combination of qualitative and quantitative principles which allow for more specific and accurate evaluation results.

The risk model of the NIMBY phenomenon of transmission and transformation projects and the evaluation method are mainly carried out in the following steps: (1) Firstly, analyze and determine the major elements affecting the NIMBY phenomenon of transmission and transformation projects, and construct a three-level evaluation index system including the target layer, unit layer and element layer. Secondly, define the evaluation grade characteristics and evaluation value ranges. Finally, estimate the risk model of the NIMBY phenomenon of transmission and transformation projects. (2) Secondly, use the expert evaluation method and radar chart method to evaluate different elements of every layer. Then, use the minimum area method to calculate the risk level. Finally, use the fuzzy comprehensive evaluation method to calculate the overall risk level. The construction of the risk model and the flow of the evaluation method are shown in Figure 1.

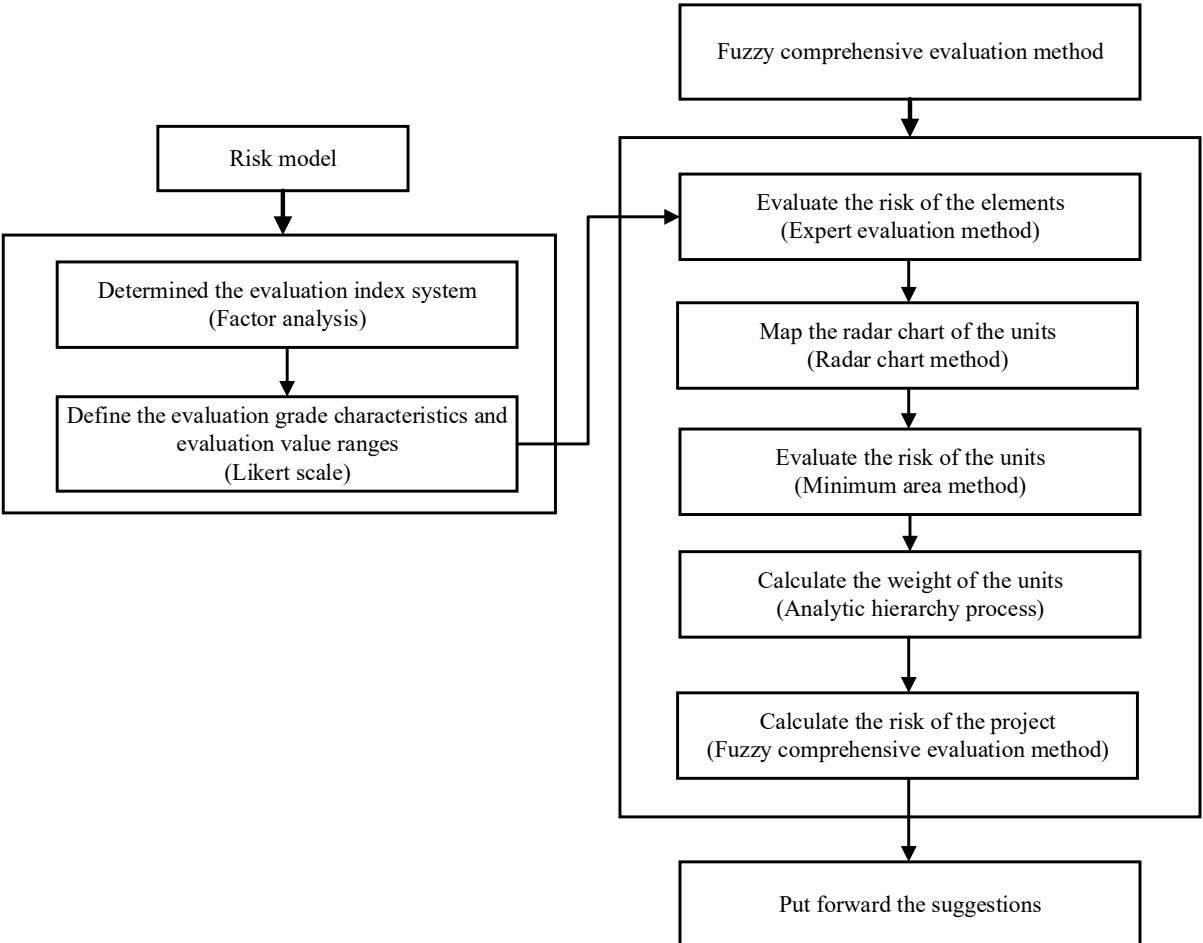

**Figure 1.** Risk model and the flow of the risk evaluation method.

A combination of multiple evaluation methods is adopted to evaluate the risk of the NIMBY phenomenon of transmission and transformation projects, making the results more accurate and reasonable. Meanwhile, a factor-analysis method is an objective and scientific mathematical method which can be used to select the main evaluation indexes of the NIMBY phenomenon and quantify the evaluation indexes through the Likert scale [39]. In addition, the expert evaluation method is scored with the help of experts, which is more professional and authoritative.

## 2.2. Evaluation Index System

The most important step of the risk model lies in the selection of evaluation indexes whose accuracy and comprehensiveness directly affect the reliability and scientificity of the evaluation results. By analyzing, extracting and summarizing the cases of transmission and transformation projects with the NIMBY phenomenon in Zhejiang Province, Chongqing Municipality, and Xi'an City, and following the laws on transmission and transformation projects formulated by China, the risk model of the NIMBY phenomenon is constructed as shown in Figure 2. Meanwhile, the risk-evaluation indexes include the power-grid corporation, the local residents, the ambiguity of relevant scientific standards and changes in national laws.

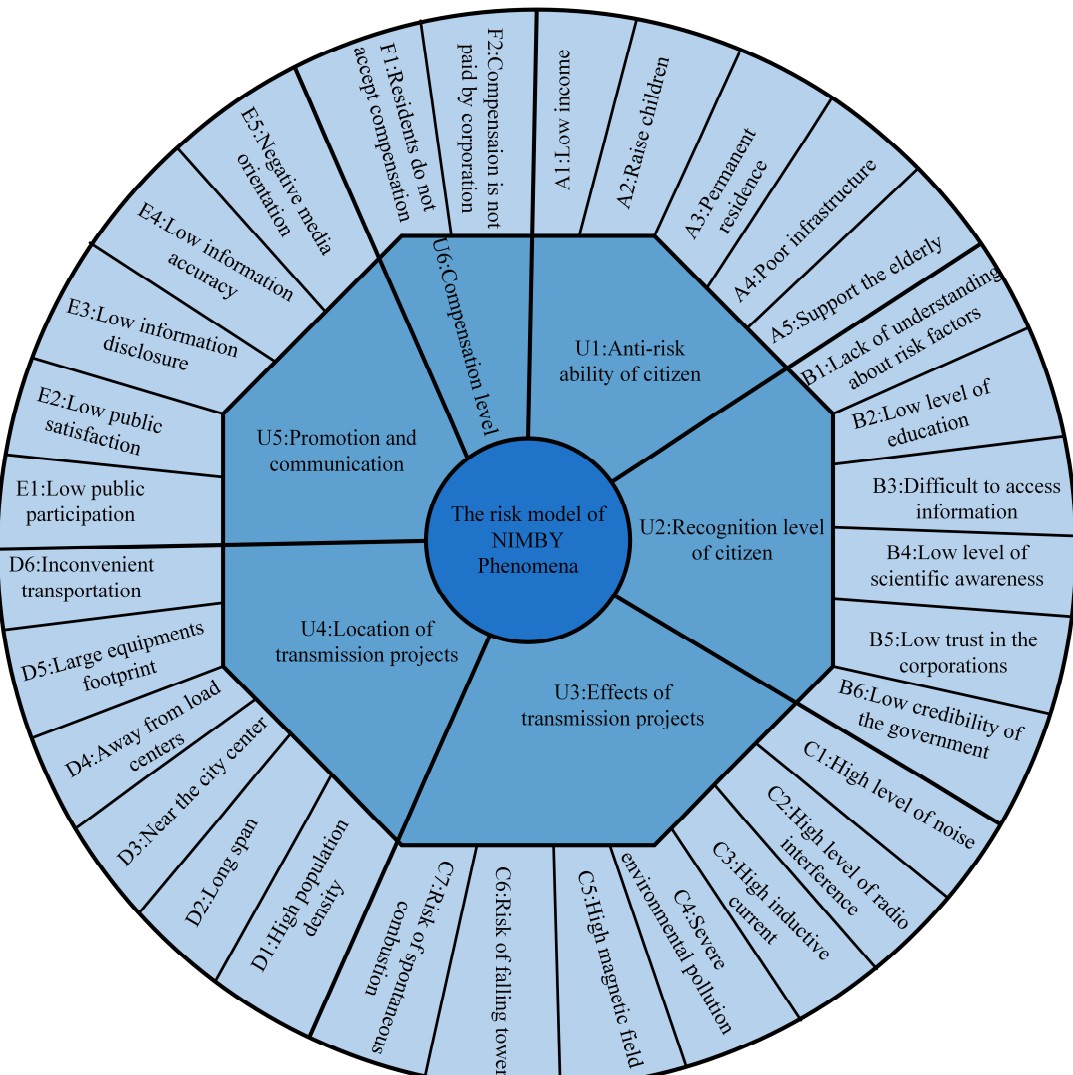

**Figure 2.** Risk model of NIMBY phenomenon.

The risk model of the NIMBY phenomenon consists of six units and thirty-one elements, as shown Figure 2. In more detail, the six units identify the main risk of the NIMBY phenomenon, and thirty-one elements point to specific influences. The subject and impact aspects of each unit are listed below [40–42]. In this figure, U1 to U6 are the abbreviations of the units. A1 to A5, B1 to B6, C1 to C7, D1 to D6, E1 to E5 and F1 to F6 are the abbreviations of the elements.

U1: Anti-risk ability of the citizen. The units focus on the resident's family situation, family income, children's status, parental status and residential property, and these factors reflect the acceptability of financial compensation to residents.

U2: Residents' understanding of transmission and transformation projects and whether they accept the construction of transmission and transformation projects.

U3: The impact of transmission and transformation projects on the lives and livelihoods of neighboring residents.

U4: Whether the location of transmission and transformation projects can reduce the probability of the NIMBY phenomenon.

U5: Whether the power-grid corporation's promotion and communication are in place to facilitate public acceptance of the construction of transmission and transformation projects.

U6: Measuring the acceptance of local residents and the power-grid corporation in economic terms.

The above units and elements are not isolated from each other, but are closely related. In the specific practice process, the elements can be adjusted appropriately according to the actual transmission and transformation project. The thirty-one elements in these units can be used as the risk-evaluation indexes of the NIMBY phenomenon in transmission and transformation projects.

### 2.3. Evaluation Grade Characteristics and Evaluation Value Ranges

Combining the cases and the laws that should be followed in the transmission and transformation projects in China, the qualitative indicator system has been obtained. However, in order to quantitatively and comprehensively evaluate the risk of the NIMBY phenomenon in transmission and transformation projects, it is necessary to clearly define the evaluation criteria for each index.

According to the research of experts and the strict laws on noise and electromagnetic environment in China, each unit and element are divided into five levels (level 1 to level 5). In more detail, the five risk levels and the range of evaluation values are uniformly corresponded to each other one by one, in which 1 and 5 points correspond to the lowest and the highest evaluation values, respectively. The risk levels and their ranges of assessed values are, in descending order, essentially no risk (0.0, 2.0], low risk (2.0, 3.2], medium risk (3.2, 4.0], high risk (4.0, 4.6] and severe risk (4.6, 5.0], as shown in Table 1.

**Table 1.** Definition of risk level.

| Risk Level | Level Definition | Range of Assessed Values/Points |
| --- | --- | --- |
| Essentially no risk | Projects have little impact on residents and are of greater benefit | (0.0, 2.0] |
| Low risk | Projects have a small impact on residents and are acceptable to most people | (2.0, 3.2] |
| Medium risk | Some aspects of projects are unreasonable and cause resistance from a small number of residents | (3.2, 4.0] |
| High risk | Most aspects of projects are unreasonable and cause resistance from most residents | (4.0, 4.6] |
| Severe risk | There is a fundamental problem with the development of projects and it needs to be planned | (4.6, 5.0] |

The key to the application of the risk-assessment results lies in the feedback on the implementation of project decisions. In the actual process of risk assessment, for projects with a medium-risk rating or higher, risk-prevention and -mitigation measures should be proposed for key risk elements, and the responsible parties and cooperation units for the measures should be clearly defined. Only after the risk of the measures has been reassessed and confirmed to be low risk can the project continue to be implemented.

## 3. Evaluation Method of the Risk Model

### 3.1. Risk Evaluation of the Unit and Element Layer

When the risk model of the NIMBY phenomenon is determined, it is crucial to conduct the risk assessment. By analyzing the flow of the risk-evaluation method, as shown in

Figure 1, it was found that the relative weights among the elements are difficult to determine because their number is large and to some extent they are interconnected and influenced by each other. Therefore, the radar charts based on the minimum-area method are used to clearly and intuitively show the risk level of each element under each unit and their relationships with each other.

### 3.1.1. Evaluation of the Risk of the Element Layer

In order to obtain statistics, relevant personnel involved in the construction of transmission and transformation projects are invited to conduct a questionnaire survey on each element based on the definition of the five risk levels and their corresponding range of evaluation values. The evaluation value can be jointly assessed by the local government, the power-grid corporation and local residents, and then the linear weighting method is used to calculate the comprehensive evaluation results. For example, the A1 is scored, and after the comprehensive calculation of the three-party evaluation, the final evaluation results are shown in Table 2. Similarly, a comprehensive evaluation of the risk model can be obtained in the end.

**Table 2.** Comprehensive evaluation results of A1.

| Investigator | Range of Assessed Values/Points | Weight |
|:---:|:---:|:---:|
| Local government | 4.4 | 0.4 |
| Power-grid corporation | 4.6 | 0.3 |
| Local residents | 4.0 | 0.3 |
| Weighted average = 4.4 × 0.4 + 4.6 × 0.3 + 4.0 × 0.3 = 4.34 | | |

### 3.1.2. Map the Radar Charts of the Units

By connecting the evaluation values calculated above by units in an $N$-sided shape, that $N$ is the number of elements corresponding to the unit layer, and a radar map for each unit can be obtained [43].

### 3.1.3. Evaluate the Risk of the Units

According to the completed radar charts, the risk of the units can be written as follows [44]:

$$r = \sqrt{\frac{S_1}{S_2}} \times 5 \tag{1}$$

where $r$ is the risk of the units, $S_1$ is the area of the shaded area, which is the area of the graph where the elements of the same unit are connected end to end, and $S_2$ is the area of the $N$-sided shape.

Since the different order of the evaluation values in the radar chart may result in different shaded areas, the risk level can be evaluated by the minimum-area-ranking method [44].

$$r = \begin{cases} \left\{ [c_1 \times (c_{2n} + c_{2n-1}) + \sum_{i=2}^{n-1} c_i \times (c_{2n-i} + c_{2n+2-i}) + c_n \times (c_{n+1} + c_{n+2})]/2n \right\}^{1/2}, N = 2n \\ \left\{ [c_1 \times (c_{2n} + c_{2n+1}) + \sum_{i=2}^{n} c_i \times (c_{2n+1-i} + c_{2n+3-i}) + c_n \times (c_{n+1} + c_{n+2})]/2(n+1) \right\}^{1/2}, N = 2n+1 \end{cases} \tag{2}$$

where $c_1, c_2, c_3, \ldots, c_{2n}, c_{2n+1}$ are in ascending order by assessed value, $i$ refers to the number of elements within the unit, $n$ represents an integer and $N$ means the total number of elements, which is divided into odd and even cases.

### 3.2. Overall Risk Evaluation

Based on the weights of each unit relative to the overall calculated above, the fuzzy comprehensive evaluation method is used to obtain the overall risk evaluation of the project.

### 3.2.1. Evaluation the Risk of the Element Layer

Through the questionnaire from the local government, the power-grid corporation and local residents, the qualitative results can be obtained. Therefore, the analytic hierarchy process is used to obtain more intuitively and accurately the risk level of the NIMBY phenomenon.

(1)    Modeling the hierarchy

The hierarchy model for assessing the risk level of the NIMBY phenomenon is divided into three layers including the goal layer, the unit layer and the element layer from inside to outside.

(2)    Building the judgment matrices

Starting at the second level of the hierarchy model, the weights of all elements in this layer against an element in the previous layer are calculated, and this calculation process is obtained by a two-by-two comparison denoted as matrix *A*. Meanwhile, the scale of matrix elements is obtained as shown in Table 3.

(3)    Calculating the weight vectors

Through the eigen decomposition of the judgment matrix *A*, the maximum eigenvalue $\lambda_{max}$ and its corresponding eigenvector *W* can be obtained. In addition, the weights of these elements are acquired by normalizing *W*.

(4)    Checking the consistency

As the weighted values are obtained through scoring by different experts, there is a possibility of self-contradiction. Therefore, it is necessary to test the consistency of these scores. The test number $C_R$ of the consistency check can be described as follows [45]:

$$C_R = \frac{\lambda_{\max} - m}{(m - 1) \times R_I} \tag{3}$$

where *m* is the number of units in the unit layer, which in this paper is 6, $R_I$ is the average stochastic consistency index corresponding to *m*, which in this paper is 1.24.

When the order is 1 and 2, the judgment matrix is always completely consistent. When the order is greater than 2, if $C_R < 0.1$, the consistency of the judgment matrix is considered acceptable; otherwise, it is necessary to adjust the judgment matrix to cause it to have satisfactory consistency [46].

**Table 3.** Saaty's scale for pairwise comparison [47].

| Saaty's Scale | The Relative Importance of the Two Sub-Elements |
|:---:|:---:|
| 1 | Equally important |
| 3 | Moderately important with one over another |
| 5 | Strongly important |
| 7 | Very strongly important |
| 9 | Extremely important |
| 2, 4, 6, 8 | Intermediate values |

### 3.2.2. Comprehensive Evaluation Values of the Risk Model

Fuzzy comprehensive evaluation models are proposed by different synthetic operations between the weight vector *A* and the single-factor evaluation matrix *R*. Table 4 shows four basic methods of common fuzzy relation synthesis operations.

**Table 4.** Mathematical model of the fuzzy comprehensive evaluation.

| Serial Number | Type | The Relative |
|---|---|---|
| 1 | Principal determinant type M ($\wedge$, $\vee$) | $\wedge$ means taking a small operation, $\vee$ means taking a large operation |
| 2 | The main factor is highlighted as I type M ($\cdot$, $\vee$) | $\cdot$ stands for an ordinary multiplication Operation, $\vee$ means taking a large operation |
| 3 | The main factor is highlighted as II type M ($\wedge$, $\oplus$) | $\wedge$ means taking a small operation, $\oplus$ means sum with an upper limit of one, namely: $x \oplus y = \min(1, x + y)$ |
| 4 | Weighted mean type M ($\cdot$, +) | $\cdot$ is a normal multiplication, and + is a normal addition |

For the same object set, different mathematical models are used to calculate the sorting results that may be different. This is in line with objective reality, as observing and analyzing the same thing from different perspectives may lead to different conclusions. Model 1, Model 2 and Model 3 are rough, while Model 4 is more accurate and suitable for a comprehensive evaluation because it considers overall factors [48].

According to the radar charts of these units, the evaluation values $R$ of the six units in the risk model can be acquired as $R_1$, $R_2$, $R_3$, $R_4$, $R_5$ and $R_6$. Therefore, the fuzzy comprehensive evaluation method can be used to calculate the risk level $L$ as follows [49]:

$$L = WR^T = (W_1, W_2, W_3, W_4, W_5, W_6)(R_1, R_2, R_3, R_4, R_5, R_6)^T \tag{4}$$

## 4. Results

### 4.1. Fuzzy Analytic Hierarchy Process (FAHP) Based on the Radar Chart Method

A substation construction project in Xi'an City required risk calculations for the NIMBY phenomenon to maximize the absence of conflicts caused by the NIMBY phenomenon and enable the project to be delivered on time. In this paper, through the construction of the risk model and the assessment of the NIMBY phenomenon, and for the high-risk factors to take relevant measures, so that the risk level is reduced to a safe range, the final project can be successfully implemented. The specific steps are as follows [50–54]:

(1) Determine the evaluation index system according to the factor-analysis method. The six units and thirty-one elements of Figure 1 are selected for the risk-evaluation index system;

(2) Define the evaluation grade characteristics and evaluation value ranges according to the Likert scale;

(3) Evaluate the risk of these elements according to the expert evaluation method and map the radar chart of these units according to the radar chart method. The local government, the power-grid corporation and the local residents are invited to conduct a questionnaire survey on each element based on the definition of the five risk levels. The statistics of the questionnaire survey including numbers and percentages for the interviews are shown in Table 5. The corresponding comprehensive scores for each element of the risk model are shown in Table 6, and the evaluation results of all elements of each unit are synthesized according to the radar chart method. For example, the linearly weighted evaluation of the seven elements from c1 to c7 of the unit U3 are 3.91, 3.22, 3.12, 3.54, 4.08, 3.26 and 3.09, respectively, and the risk evaluation of the unit U3 gives 3.42 according to Equation (2). Since the number of elements in the unit U6 is less than four, the risk evaluation of the unit U6 is acquired by a weighted average. Similarly, the risk evaluation of the unit from U1 to U6 are 4.11, 4.15, 3.42, 3.90, 4.12 and 2.91. The radar charts of the units are written as shown in Figure 3;

(4) Calculate the overall risk evaluation of the transmission and transformation project according to the fuzzy comprehensive evaluation method as shown in Section 3.2. After obtaining the weights of each unit and element, the judgement matrix *A* can be ac-

quired through expert scoring of the study of the NIMBY phenomenon in transmission and transformation projects. The judgement matrix $A$ is written as follows:

$$A = \begin{bmatrix} 1 & 2 & 4 & 4 & 2 & 1/2 \\ 1/2 & 1 & 3 & 3 & 1 & 1/3 \\ 1/4 & 1/3 & 1 & 1 & 1/3 & 1/5 \\ 1/4 & 1/3 & 1 & 1 & 1/3 & 1/5 \\ 1/2 & 1 & 3 & 3 & 1 & 1/3 \\ 2 & 3 & 5 & 5 & 3 & 1 \end{bmatrix} \tag{5}$$

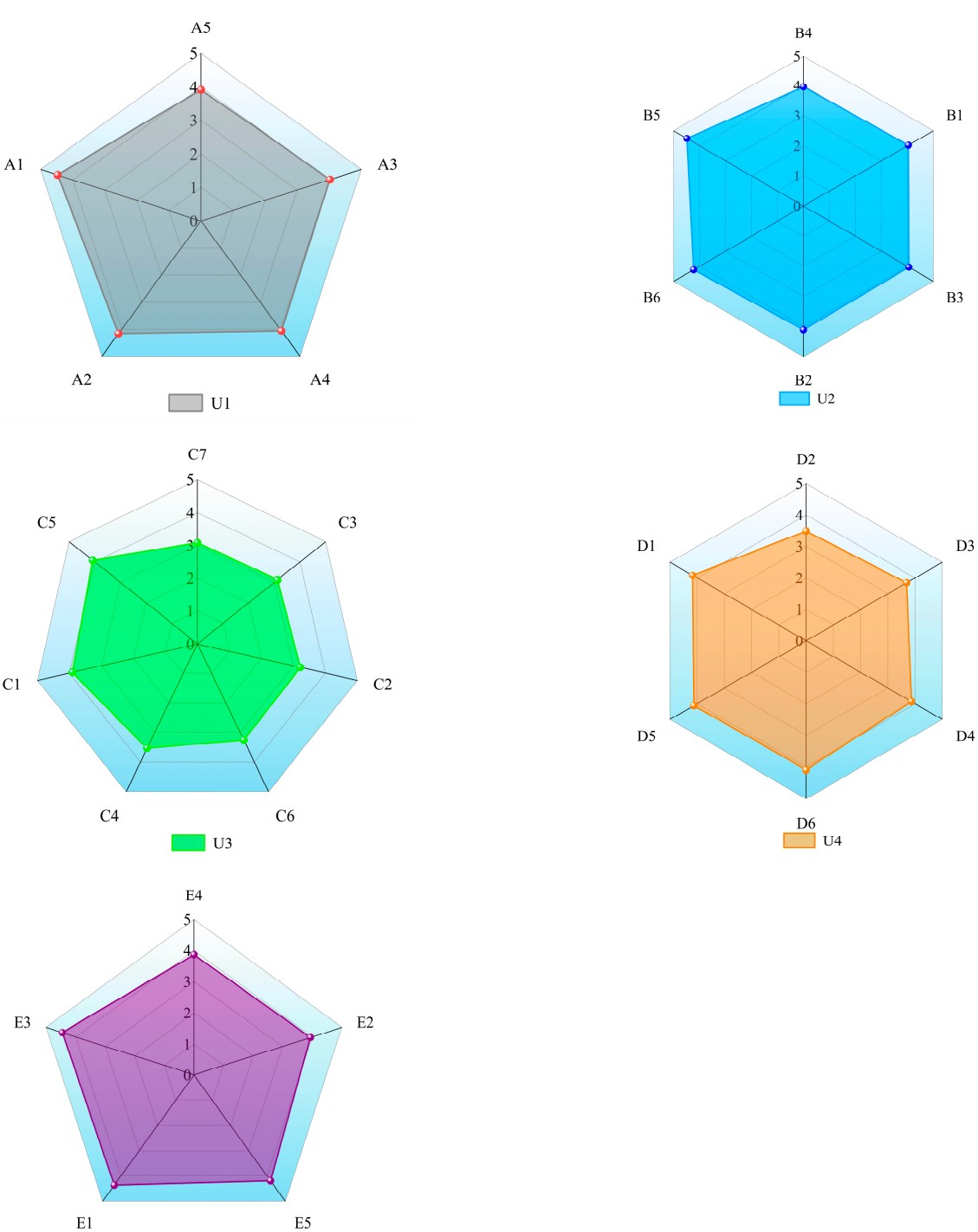

**Figure 3.** Radar charts of the units from U1 to U5.

**Table 5.** The statistics of the questionnaire survey.

| Investigator | Number | Percentage |
|---|---|---|
| Local government | 33 | 21.7% |
| Power-grid corporation | 48 | 31.6% |
| Local residents | 71 | 46.7% |
| Total | 152 | 100% |

**Table 6.** Comprehensive score for each element of the risk model.

| Units | Elements | Comprehensive Score |
|---|---|---|
| U1: Anti-risk ability of citizens | A1: Low income | 4.47 |
| | A2: Raise children | 4.16 |
| | A3: Permanent residence | 4.02 |
| | A4: Poor infrastructure | 4.06 |
| | A5: Support the elderly | 3.92 |
| U2: Recognition level of citizens | B1: Lack of understanding about risk factors | 4.04 |
| | B2: Low level of education | 4.11 |
| | B3: Difficult to access information | 4.06 |
| | B4: Low level of scientific awareness | 3.96 |
| | B5: Low trust in the corporations | 4.49 |
| | B6: Low credibility of the government | 4.23 |
| U3: Effects of transmission and transformation projects | C1: High level of noise | 3.91 |
| | C2: High level of radio interference | 3.22 |
| | C3: High inductive current | 3.12 |
| | C4: Severe environmental pollution | 3.54 |
| | C5: High magnetic field | 4.08 |
| | C6: Risk of falling tower | 3.26 |
| | C7: Risk of spontaneous combustion | 3.09 |
| U4: Location of transmission and transformation projects | D1: High population density | 4.17 |
| | D2: Long span | 3.49 |
| | D3: Near the city center | 3.69 |
| | D4: Away from load centers | 3.87 |
| | D5: Large equipment footprint | 4.12 |
| | D6: Inconvenient transportation | 4.09 |
| U5: Promotion and communication | E1: Low public participation | 4.36 |
| | E2: Low public satisfaction | 3.94 |
| | E3: Low information disclosure | 4.45 |
| | E4: Low information accuracy | 3.87 |
| | E5: Negative media orientation | 4.19 |
| U6: Compensation level | F1: Residents do not accept compensation | 3.09 |
| | F2: Compensation is not paid by corporation | 2.73 |

The maximum eigenvalue $\lambda_{max}$ is 6.074 and its corresponding normalized eigenvector $W$ = [0.2349, 0.1425, 0.0564, 0.0564, 0.1425, 0.3675] is obtained. In addition, the test number $C_R$ is 0.012, which is less than 0.1, so the judgement matrix $A$ is valid because it passes the consistency check.

Therefore, the risk evaluation of the NIMBY phenomenon obtained is 3.3530, proving that the risk level is medium risk according to Table 1.

### 4.2. Fuzzy Analytic Hierarchy Process (FAHP)

The method utilizes the four-step FAHP method to determine the weights of the criteria. Experts are required to provide their judgments on the basis of their expertise and to compare every factor pairwise in their corresponding section structured in the hierarchy. Then, these experts' preferences are converted into a fuzzy number-evaluation matrix. The defuzzication is employed to transform the fuzzy scales into crisp scales for the computation of priority weights.

The judgement matrix *A* of units can be acquired using Equation (5) through the expert scoring of the study. Similarly, the judgement matrix *A* of elements can be calculated as follows:

$$U_1 = \begin{bmatrix} 1 & 1/2 & 1/3 & 1/5 & 1/2 \\ 2 & 1 & 1/2 & 1/4 & 1 \\ 3 & 2 & 1 & 1/2 & 2 \\ 5 & 4 & 2 & 1 & 3 \\ 2 & 1 & 1/2 & 1/3 & 1 \end{bmatrix} \tag{6}$$

$$U_2 = \begin{bmatrix} 1 & 3 & 2 & 3 & 5 & 4 \\ 1/3 & 1 & 1/3 & 1/2 & 3 & 2 \\ 1/2 & 3 & 1 & 2 & 4 & 3 \\ 1/3 & 2 & 1/2 & 1 & 3 & 2 \\ 1/5 & 1/3 & 1/4 & 1 & 3 & 2 \\ 1/4 & 1/2 & 1/3 & 1/2 & 2 & 1 \end{bmatrix} \tag{7}$$

$$U_3 = \begin{bmatrix} 1 & 1 & 1 & 1/2 & 1 & 1/2 & 1/2 \\ 1 & 1 & 1 & 2 & 1 & 1/2 & 1/2 \\ 1 & 1 & 1 & 2 & 1 & 1/2 & 1/2 \\ 2 & 1/2 & 1/2 & 1 & 1/2 & 1/3 & 1/3 \\ 1 & 1 & 1 & 2 & 1 & 1/2 & 1/2 \\ 2 & 2 & 2 & 3 & 2 & 1 & 1 \\ 2 & 2 & 2 & 3 & 2 & 1 & 1 \end{bmatrix} \tag{8}$$

$$U_4 = \begin{bmatrix} 1 & 1/4 & 1/2 & 1/3 & 1/4 & 1/5 \\ 4 & 1 & 3 & 2 & 1 & 1/2 \\ 2 & 1/3 & 1 & 1/2 & 1/3 & 1/4 \\ 3 & 1/2 & 2 & 1 & 1/2 & 1/3 \\ 4 & 1 & 3 & 2 & 1 & 1/2 \\ 5 & 2 & 4 & 3 & 2 & 1 \end{bmatrix} \tag{9}$$

$$U_5 = \begin{bmatrix} 1 & 2 & 1 & 2 & 3 \\ 1/2 & 1 & 1/2 & 1 & 2 \\ 1 & 2 & 1 & 2 & 3 \\ 1/2 & 1 & 1/2 & 1 & 2 \\ 1/3 & 1/2 & 1/3 & 1/2 & 1 \end{bmatrix} \tag{10}$$

$$U_6 = \begin{bmatrix} 1 & 3 \\ 1/3 & 1 \end{bmatrix} \tag{11}$$

The weights of the 31 indicators are $W_1$ = [0.0734, 0.1259, 0.2365, 0.4309, 0.1332], $W_2$ = [0.3578, 0.1166, 0.2429, 0.1512, 0.0513, 0.0802], $W_3$ = [0.1030, 0.1188, 0.1188, 0.0863, 0.1188, 0.2271, 0.2271], $W_4$ = [0.0496, 0.2054, 0.0762, 0.1226, 0.2054, 0.3409] and $W_5$ = [0.2976, 0.1579, 0.2976, 0.1579, 0.0890], $W_6$ = [0.7500, 0.2500].

The weights of the 31 evaluation indicators were calculated using the FAHP method. Based on the collected survey results, the weights of each indicator were calculated and are presented in Table 7. The risk evaluation of the NIMBY phenomenon obtained is 3.6555, proving that the risk level is medium risk according to Table 1.

**Table 7.** The weights of each indicator.

| Target Layer | Primary Index | $w_i$ | Secondary Index | $w_i$ | $w_{ij}$ | Ranking |
|---|---|---|---|---|---|---|
| The risk evaluation | U1: Anti-risk ability of citizens | 0.2349 | A1: Low income | 0.0734 | 0.0172 | 15 |
| | | | A2: Raise children | 0.1259 | 0.0296 | 10 |
| | | | A3: Permanent residence | 0.2365 | 0.0556 | 4 |
| | | | A4: Poor infrastructure | 0.4309 | 0.1012 | 2 |
| | | | A5: Support the elderly | 0.1332 | 0.0313 | 9 |

**Table 7.** *Cont.*

| Target Layer | Primary Index | $w_i$ | Secondary Index | $w_i$ | $w_{ij}$ | Ranking |
|---|---|---|---|---|---|---|
| The risk evaluation | U2: Recognition level of citizens | 0.1425 | B1: Lack of understanding about risk factors | 0.3578 | 0.0510 | 5 |
| | | | B2: Low level of education | 0.1166 | 0.0166 | 16 |
| | | | B3: Difficult to access information | 0.2429 | 0.0346 | 8 |
| | | | B4: Low level of scientific awareness | 0.1512 | 0.0215 | 13 |
| | | | B5: Low trust in the corporations | 0.0513 | 0.0073 | 23 |
| | | | B6: Low credibility of the government | 0.0802 | 0.0114 | 22 |
| | U3: Effects of transmission and transformation projects | 0.0564 | C1: High level of noise | 0.1030 | 0.0058 | 28 |
| | | | C2: High level of radio interference | 0.1188 | 0.0067 | 25 |
| | | | C3: High inductive current | 0.1188 | 0.0067 | 25 |
| | | | C4: Severe environmental pollution | 0.0863 | 0.0049 | 29 |
| | | | C5: High magnetic field | 0.1188 | 0.0067 | 25 |
| | | | C6: Risk of falling tower | 0.2271 | 0.0128 | 17 |
| | | | C7: Risk of spontaneous combustion | 0.2271 | 0.0128 | 17 |
| | U4: Location of transmission and transformation projects | 0.0564 | D1: High population density | 0.0496 | 0.0028 | 31 |
| | | | D2: Long span | 0.2054 | 0.0116 | 20 |
| | | | D3: Near the city center | 0.0762 | 0.0043 | 30 |
| | | | D4: Away from load centers | 0.1226 | 0.0069 | 24 |
| | | | D5: Large equipment footprint | 0.2054 | 0.0116 | 20 |
| | | | D6: Inconvenient transportation | 0.3409 | 0.0192 | 14 |
| | U5: Promotion and communication | 0.1425 | E1: Low public participation | 0.2976 | 0.0424 | 6 |
| | | | E2: Low public satisfaction | 0.1579 | 0.0225 | 11 |
| | | | E3: Low information disclosure | 0.2976 | 0.0424 | 6 |
| | | | E4: Low information accuracy | 0.1579 | 0.0225 | 11 |
| | | | E5: Negative media orientation | 0.0890 | 0.0127 | 19 |
| | U6: Compensation level | 0.3675 | F1: Residents do not accept compensation | 0.7500 | 0.2756 | 1 |
| | | | F2: Compensation is not paid by corporation | 0.2500 | 0.0919 | 3 |

The above methods obtain the same level of safety, but considering that the radar chart method considers more comprehensive factors, it is better to be comprehensive.

## 5. Discussion

Based on the experience of relevant transmission and substation projects, it is concluded that a medium risk is highly likely to have conflicts caused by the NIMBY phenomenon and that there is huge scope for the all-round enhancement of the project. Therefore, we need to analyze the risk level of each unit of the project. Through the analysis, we can establish that U1, U2 and U5 are at high risk, U3 and U4 are at medium risk and U6 is at low risk, which shows that the public's ability to resist risk is weak, and the level of public acceptance and publicity is not in place in the location of the project, so the amount of compensation should be increased appropriately, increasing the publicity of the impact of transmission and transformation projects and public participation. In addition, the medium risk shows that there is also a risk of impact from the project facilities themselves, and based on onsite measurements, it was found that the noise and magnetic field strengths are indeed high. Therefore, treatment is needed to reduce the impact of the project itself, so that the risk index meets the requirements.

According to whether the evaluation value varies with the measures taken, units can be divided into two categories. The first type is U1 and U4. U1 reflects the standard of living of the residents, and it does not change with the situation of the site, nor with the measures

taken. U4 represents the characteristics of the power transmission and transformation project itself, which changes with the scale of the project.

The second type is U2, U3, U5 and U6. U2 reports the level of citizens' awareness of science and the NIMBY phenomenon. When hearings and discussion meetings are held to make residents fully aware of the huge benefits and overestimated disadvantages of power transmission and transformation projects, the assessment value of the unit will be greatly reduced. U3 reflects the impact of power transmission and transformation projects, which is not only physical but also psychological. In response to noise and other physical impacts, relevant departments should listen to public opinions and take measures promptly. Meanwhile, scientific lectures should be held to enhance the public's correct understanding of risks and reduce their psychological impact. U5 reports whether the publicity of the government and power-grid corporation is in place. If the evaluation value of this unit is too high, it is necessary to rectify the publicity department. U6 reflects the conflict point between the public and power-grid corporations. When the public's living conditions are poor, the compensation amount can be appropriately increased. After receiving scientific learning, the public will also reduce their hostility towards the NIMBY phenomenon. Therefore, there are certain lower limits to the risk evaluation in different regions. In addition, the purpose of this paper is to take measures for high-risk units and minimize the overall risk.

After taking the above measures, a second questionnaire survey was conducted on previous interviewers. Meanwhile, the content of this questionnaire was the same as that of the first one, but some of the interviewers were unable to accept the invitation. Therefore, the corresponding number of investigators was supplemented, as shown in Table 5. The corresponding comprehensive scores for each element of the risk model are shown in Table 8.

**Table 8.** Comprehensive score after taking measures for each element of the risk model.

| Units | Elements | Comprehensive Score |
|---|---|---|
| U1: Anti-risk ability of citizens | A1: Low income | 4.42 |
| | A2: Raise children | 4.46 |
| | A3: Permanent residence | 4.35 |
| | A4: Poor infrastructure | 4.05 |
| | A5: Support the elderly | 4.07 |
| U2: Recognition level of citizens | B1: Lack of understanding about risk factors | 2.84 |
| | B2: Low level of education | 4.05 |
| | B3: Difficult to access information | 2.43 |
| | B4: Low level of scientific awareness | 2.50 |
| | B5: Low trust in the corporations | 2.51 |
| | B6: Low credibility of the government | 2.47 |
| U3: Effects of transmission and transformation projects | C1: High level of noise | 2.97 |
| | C2: High level of radio interference | 2.34 |
| | C3: High inductive current | 2.52 |
| | C4: Severe environmental pollution | 2.71 |
| | C5: High magnetic field | 2.44 |
| | C6: Risk of falling tower | 2.24 |
| | C7: Risk of spontaneous combustion | 2.35 |
| U4: Location of transmission and transformation projects | D1: High population density | 4.11 |
| | D2: Long span | 3.52 |
| | D3: Near the city center | 3.51 |
| | D4: Away from load centers | 3.74 |
| | D5: Large equipment footprint | 3.67 |
| | D6: Inconvenient transportation | 3.91 |
| U5: Promotion and communication | E1: Low public participation | 2.23 |
| | E2: Low public satisfaction | 2.25 |
| | E3: Low information disclosure | 2.74 |
| | E4: Low information accuracy | 2.36 |
| | E5: Negative media orientation | 2.17 |
| U6: Compensation level | F1: Residents do not accept compensation | 2.23 |
| | F2: Compensation is not paid by corporation | 2.51 |

Through the use of the radar chart method based on the minimum-area method, the risk evaluations of the unit from U1 to U6 are 4.23, 2.78, 2.51, 3.74, 2.32 and 2.37. By comparing the statistics before and after taking measures, two conclusions can be drawn which are also in line with the fact. On the one hand, the risk evaluation of U1 and U4 remains basically unchanged because the living conditions and location of the residents have not changed significantly. On the other hand, the evaluation values of other units have shown varying degrees of decrease, because through holding hearings and discussion meetings, the public has a clearer understanding of the advantages of power transmission and transformation projects, which greatly reduces the evaluation of units and the risk of the NIMBY phenomenon.

In addition, the corresponding normalized eigenvector $W$ remains unchanged and the risk evaluation of the NIMBY phenomenon obtained is reduced to 2.93 according to Equation (5), which already meets the construction requirements, and finally the project is successfully implemented in Shaanxi Province without generating conflicts caused by the NIMBY phenomenon.

It is essential to calculate the risk evaluation of the NIMBY phenomenon, which we insist must be assessed before the construction of the transmission station project, causing the risk level of the NIMBY phenomenon to be reduced to low-risk, or else a huge amount of time and money will be wasted.

## 6. Conclusions

In this paper, a risk evaluation model of the NIMBY phenomenon was established using the fuzzy comprehensive evaluation method, the radar chart method and the minimum-area method. Combining the cases and the laws that should be followed in the transmission and transformation projects in China, a total of 31 parameters were selected as evaluation indicators, forming a risk-evaluation system and model of the NIMBY phenomenon. In addition, the scores corresponding to these criteria are obtained by questionnaire surveys. The radar chart method and the minimum area method are applied to determine the weights of the element and unit layers. Furthermore, the overall risk level is evaluated by the fuzzy comprehensive evaluation method. Finally, a transmission and transformation project in Xi'an City, China, is used as an example to verify the correction of the risk model and its evaluation method. The results show that based on the risk model and fuzzy comprehensive evaluation method, the limitations of the signal evaluation method on the risk of the NIMBY phenomenon can be avoided, and the results shows that the proposed method is feasible and effective.

The contribution of this paper is to propose a risk-evaluation model that assesses the risk level of the NIMBY phenomenon before the construction of power transmission and transformation projects. By using this evaluation model to assess the risk level, the weakness can be analyzed and suggestions for decreasing the risk level are put forward to minimize losses due to the NIMBY phenomenon.

However, some limitations can be expected. The risk evaluation of the NIMBY phenomenon is an extremely time-consuming and complex task that involves a wide range of knowledge. The proposed method does not consider all possible criteria that can be added to the risk model which includes human factors such as differences among different interviewers. In addition, the risk level may change if a new criterion is added.

For future work, we intend to explore more cases and conduct more empirical studies to further validate the usefulness of the research.

**Author Contributions:** J.W. proposed the topic of the review. Z.W. surveyed the literature and composed the manuscript. Z.W. conducted the literature review. N.D., J.W. and X.B. discussed and revised the manuscript. N.D. and J.W. supervised this project. J.W. and Z.W. contributed equally to this work. All authors have read and agreed to the published version of the manuscript.

**Funding:** This research was funded in part by the National Natural Science Foundation of China under Grant 52077161.

**Institutional Review Board Statement:** Not applicable.

**Informed Consent Statement:** Not applicable.

**Data Availability Statement:** The original contributions presented in the study are included in the article, further inquiries can be directed to the corresponding author.

**Conflicts of Interest:** Authors Jian Wu and Xiaochun Bai were employed by the company Shaanxi Electric Power Research Institute of State Grid Corporation of China. The remaining authors declare that the research was conducted in the absence of any commercial or financial relationships that could be construed as a potential conflict of interest.

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
