# Peer review of "Comprehensive Evaluation of NIMBY Phenomenon with Fuzzy Analytic Hierarchy Process and Radar Chart"

_applsci, doi:10.3390/app14062654_

Round 1
Reviewer 1 Report
Comments and Suggestions for Authors
Comments
General Comments
This is a study dealing with a very interesting subject, related with the NIMBY Phenomenon of power transmission and transformation project by utilizing Fuzzy Analytic Hierarchy Process and Radar Chart. This manuscript could be a nice contribution to Applied Sciences journal, however some issues should be resolved first. For example, I have noticed that the article isn’t following the standard structure of a manuscript (introduction, methodology, results, and discussion parts). So, I recommend to authors to apply this in their manuscript, since it would help the presentation of the content.
Specific Comments
· Line 10: please explain NIMBY term here.
· Line 10: please be more specific. Transmission and transformation project of what?
· Line 28-29: “The electric power industry is the most important basic energy industry in the development of national economy” please provide a source for this statement or revise the sentence.
· Line 33-34: “reducing risk assessment levels for the electric power industry helps to reduce construction costs” a reference for this part is recommended.
· Line 92: DMs?
· Line 95-96: " Laarhoven and Pedrycz” please cite the mentioned reference.
· Line 97: “Sachin et al.” use the reference number appropriately not at the end of the sentence. Please make sure to apply this for the rest of the text.
· Line 128: please emphasize here why this method is more comprehensive and reliable
· Line 152-153: “factor analysis method is an objective and scientific mathematical method, which can be used to select the main evaluation indexes” Why? please briefly justify this statement and a relevant reference.
· Line 252 & 256: please provide relevant references for equation 1 & equation 2. Same for the rest equations
Author Response
Please see the attachment, thank you very much for this review.

Reviewer 2 Report
Comments and Suggestions for Authors
This paper is about NIMBY phenomenom specially focus on transmission and transformation projects, by using fuzzy analytic hierarchy process and radar chart methods for quantifying.
Is highly recommended for authors to focus more about the design of analysis of electromagnetic devices as transmision and transformation technological projects, not only on the social impact of it.
Section 4 have some issues that need to be clarified, first section 4.1 repeat the word elements at least 12 times. Equation 1 needs to be explained which is the "shaded" area, o may be is the enclosed area by the vertices, and/or figure 3 needs to shade the area of each radar chart. Equation 2 also needs to clarify the definition of n, i and N.
Section 5 explain the application of the two methods in one specific case, the questionnaire survey was conducted by the authors? How many people of each group were invited to applied? Both methods give almost the same risk evaluation of NIMBY phenomenon number, but at the end of the section authors said that by "taking relevant measures" this number is reduced to 1.93. How is that possible?, those measures and what authors do to reduce this number is of high relevance for the study and the paper, is highly recommended to give more information about it.
Finally, conclusions are too weak, authors said that the risk of NIMBY phenomenon can be avoided and the results can be comprehensively, systematically and accurately evaluated and use it to reduce the risk level due to NIMBY phenomenom, but no data is given about how this can be done, the only case that is studied, gives medium risk level, and authors reduce it but no data or explanation about how was reduced were given by them. Please clarify it, and is better if you have more than one study to support these conclusions.
Comments on the Quality of English Language
Check some grammar issues, for example, try not to repeat the same word so many times.
Author Response

(The authors gave the same response as above.)

Reviewer 3 Report
Comments and Suggestions for Authors
The paper addresses an important issue - the NIMBY phenomenon in transmission and transformation projects - which has significant implications for safety, economy, and public acceptance. The paper proposes a comprehensive evaluation method that considers thirty-one evaluation criteria and utilizes both qualitative and quantitative analysis techniques, including the fuzzy analytic hierarchy process (FAHP), radar chart method, and fuzzy comprehensive evaluation method. The use of a real-world example, the transmission and transformation project in Xi’an City, China, adds practical relevance and helps validate the proposed method.
Areas for Improvement:
1. The text lacks clarity in some parts and could benefit from improved organization. For example, the transition between sections could be smoother to enhance readability.
2. While the paper provides a general overview of the NIMBY phenomenon and mentions some relevant research, a more thorough literature review would strengthen the theoretical foundation of the study and demonstrate a deeper understanding of the existing knowledge in the field.
3. The methods employed (FAHP, radar chart method, etc.) are briefly described, but more detailed explanations and justifications are needed to ensure the rigor and reproducibility of the research. This includes clarity on how the criteria were selected, the rationale behind choosing specific methods, and how the weights were determined.
4. The paper mentions the use of a questionnaire survey to obtain scores for evaluation criteria, but details regarding the survey design, sample size, representativeness, and statistical analysis are lacking. Providing more information on these aspects would enhance the credibility of the findings.
5. The paper claims to offer novel contributions, but these need to be articulated more clearly. Specifically, the paper should clearly outline how the proposed method advances existing approaches and fills gaps in the literature.
Author Response

(The authors gave the same response as above.)

Round 2
Reviewer 2 Report
Comments and Suggestions for Authors
Althought this article had been reviewed by authors, the main focus of it seems to be the social impact not the design of analysis of electromagnetic devices as transmission and transformation technological projects.
In the article authors said that the risk evaluation of NIMBY phenomenon is 3.3 and 3.6, then by appliying a new survey this is reduced to 2.93, but also said that different persons participated in this new interview. So this difference can be due to the new participants or which specific actions were performed to change the perception of the interviewers, just by holding hearings and discussion meetings this risk reduces?
Only one survey is used to concluded that the proposed method is useful, more evidences are needed to fully concluded that this method was useful and no other factors change the risk evaluation number, is highly recommended to use more examples or experiments to conclude it.
Comments on the Quality of English LanguageNo comments
Author Response
Please see the attachment, thank you very much for your second review.

Reviewer 3 Report
Comments and Suggestions for Authors
It can be accepted with this version.
Author Response
Thank you very much for your postive review.
Round 3
Reviewer 2 Report
Comments and Suggestions for Authors
No more comments